



**Soil Atterberg limits of different weathering profiles of the collapsing**
**gullies in the hilly granitic region of south China**
**Yusong Deng [1], Chongfa Cai [1]\*, Dong Xia [2], Shuwen Ding [1], Jiazhou Chen [1]**
[1]*Key Laboratory of Arable Land Conservation (Middle and Lower Reaches of Yangtze River) of the Ministry of*
*Agriculture, College of Resources and Environment, Huazhong Agricultural University, Wuhan, 430070,*
*People's Republic of China*
[2] *College of hydraulic and Environmental engineering, China Three Gorges University, Yichang 443002, China*
\*Corresponding author. E-mail: chongfacai@126.com
Post address:
College of Resources and Environment, Huazhong Agricultural University, Wuhan 430070, China.
Tel: +86-27-87288249
Fax: +86-27-87288249
**Other co-authors' e-mails:**   dennyus@163.com (Yusong Deng)
xiadongsanxia@163.com (Dong Xia)
dingshuwen@mail.hzau.edu.cn (Shuwen Ding)
jzchen@mail.hzau.edu.cn (Jiazhou Chen)





**Abstract.** Collapsing gully erosion is one of the most serious natural hazards in the hilly granitic
region of south China. However, few studies have been performed on the relationship of soil
Atterberg limits with soil profiles of the collapsing gullies. Soil Atterberg limits, which include
plastic limit and liquid limit, have been proposed as indicators for soil vulnerability to degradation.
Here, the soil Atterberg limits within different weathering profiles and their relationships with soil
physico-chemical properties were investigated by characterizing four collapsing gullies in four
counties (Tongcheng County, Gan County, Anxi County and Wuhua County, labeled as TC, GX,
AX and WH, respectively) in the hilly granitic region of southern China. The results showed that
with the fall of weathering degree (from surface layer to detritus layer), there was a sharp decrease
in plastic limit, liquid limit, plasticity index, soil organic matter, cation exchange capacity and free
iron oxide, a gradual increase in liquidity index, a sharp increase in particle density and bulk density
followed by a slight decline, as well as a decrease in the finer soil particles (silt and clay), a
noticeable decline in the clay contents, and a considerable increase in the gravel and sand contents.
The plastic limit varied from 19.43 to 35.93 % in TC, 19.51 to 33.82 % in GX, 19.32 to 35.58 % in
AX and 18.91 to 36.56 % in WH while the liquid limit varied from 30.91 to 62.68 % in TC, 30.89
to 57.70 % in GX, 32.48 to 65.71 % in AX and 30.77 to 62.70 % in WH, respectively. The soil
Atterberg limits in the sandy soil layers and detritus layers were lower than those in the surface
layers and red soil layers, leading to the loss of bottom soil layers, the collapse of upper soil layers
and finally the occurrence of collapsing gully erosion. The regression equation showed that soil
Atterberg limits had significant and positive correlation with SOM, clay content, CEC and $Fe_d$,
significant and negative correlation with sand content and no obvious correlation with other
properties. The results of this study revealed that soil Atterberg limits are an informative indicator
to reflect the weathering degree of different weathering profiles of the collapsing gullies in the hilly
granitic region.
**1    Introduction**
In 1911, Atterberg proposed the limits of consistency for agricultural purposes to get a clear
concept of the range of water contents of a soil in the plastic state (Atterberg, 1911). These limits
of consistency, namely plastic limit and liquid limit, are well known as soil Atterberg limits. Plastic





limit is the boundary between semi-solid and plastic state, and liquid limit separates plastic state
from liquid state (Campbell, 2001). The methods developed by Casagrande (1932, 1958) to
determine the liquid and plastic limits are considered as standard international tests. The width of
the plastic state (liquid limit minus plastic limit), the plasticity index, is very useful for
characterization, classification and prediction of the engineering behavior of fine soils. Moreover,
some research attempts have been made on the relationship between in situ water content and
Atterberg limits, the liquidity index, which is the ratio of the difference between the natural moisture
content and the plastic limit to the plastic limit (Intan et al., 2014; Rashid et al., 2014). Atterberg
limits were used in early studies on the tillage of soils, with the plastic limit recognized as the highest
possible soil water content for cultivation (Baver, 1930; Jong et al., 1990). Later on, Atterberg limits
were mainly used in the classification of soils for engineering purposes. They also provide
information for interpreting several soil mechanical and physical properties such as shear strength,
bearing capacity, compressibility and shrinkage-swelling potential (Archer, 1975; Wroth, 1978;
Cathy et al., 2008; McBride, 2008). Meanwhile, Atterberg limits are also essential for infrastructure
design (e.g., construction of buildings and roads) (Zolfaghari et al., 2015). These studies clearly
show that there is a close relationship between Atterberg limits and certain properties of soils. More
recently, Atterberg limits have been proposed as indicators for soil vulnerability to degradation
processes of both natural and anthropogenic origin. Yalcin (2007) emphasized that, when subjected
to water saturation, soils with limited cohesion are susceptible to erosion during heavy rainfall.
Curtaz et al. (2014), Vacchiano et al. (2014) and Stanchi et al. (2012) provided a novel overview on
plastic limit and liquid limit in common soil types and proposed plastic limit and liquid limit as
indicators to assess the vulnerability of mountain soils to erosion.

Soil erosion is important problems in mountain areas as remarked by Douglas et al. (2011) and

MorenoRamón et al. (2014), and may result in considerable soil degradation (Cerdà et al., 2007;
Pavlova et al., 2014; Jordán et al., 2014; Peng et al., 2015; Muñoz-Rojas et al., 2016). Collapsing
gully is a serious type of soil erosion widely distributed in the hilly granitic region of southern
China, which is formed in the hill slopes covered by thick granite weathering mantle (Xu, 1996).
Collapsing gully was first proposed by Zeng in 1960, which is a composite erosion formed by
hydraulic scour and gravitational collapse (Zeng, 1960; Jiang et al., 2014; Xia et al., 2015; Deng et





al., 2016). These gullies develop quickly and erupt suddenly, with an annual average erosion of
over 50 kt km$^{-2}$ yr$^{-1}$ in these areas, more than 50-fold faster than the erosion on gentler slopes or
on slopes with high vegetation cover (Zhong et al., 2013). The flooding, debris flows, and other
disasters resulting from collapsing gullies can jeopardize sustainable development in the related
regions. From 1950 to 2005, gully erosion affected 1220 km$^2$ in the granitic red clay soil region,
leading to the loss of more than 60 Mt of soil (Zhang, 2010). It is worth mentioning that the
collapsing gullies in turn caused the loss of 360,000 ha of farmland, 521,000 houses, 36,000 km of
road, 10,000 bridges, 9000 reservoirs, and 73,000 ponds, as well as an economic loss of 3.28
billion USD that affected 9.17 million residents (Jiang et al., 2014; Liang et al., 2009). According
to a 2005 survey by the Monitoring Center of Soil and Water Conservation of China, collapsing
gullies are widely distributed in the granitic red clay soil regions of south China, which includes
Guangdong, Jiangxi, Hubei, Hunan, Fujian, Anhui, and Guangxi provinces. It is incredible that the
number of collapsing gullies is up to 239, 100, posing a serious threat to the local people (Feng et
al., 2009). A collapsing gully system consists of five parts: (1) upper catchment, where a large
amount of water is accumulated; (2) collapsing wall, where mass soil wasting, water erosion and
gravity erosion are quite serious; (3) colluvial deposit, where residual material is deposited; (4)
scour channel, where the sediment accumulation and transport is usually significantly deep and
narrow; (5) alluvial fan, the zone below the gully mouth where sediments transported by the
collapse are deposited (Xu, 1996; Sheng and Liao, 1997; Xia et al., 2015) (Figure 1). Collapsing
gully poses a serious problem for land utilization and development and the establishment of
sustainable environmental solutions in southern China. Unfortunately, there is no effective
approach to prevent such disasters currently, and this soil erosion has affected the lives of tens of
millions of Chinese citizens (Gao et al., 2011).

In a collapsing gully system, slumps and massive collapses of the collapsing wall are one of the

main influential factors causing the collapsing gully enlargement and development (Xia et al.,
2015). Researchers have paid close attention to the damage of collapsing gully, and found that
there is a close relationship among the stability of the collapsing wall, the erosion amount and the
development speed (Xu, 1996; Sheng and Liao, 1997; Luk et al., 1997a, 1997b; Lan et al., 2003).
Qiu (1994) pointed out that the mechanical composition of soil and the change of its action with



water have an important influence on the development of collapsing gully. Li (1992) stated that
there is an important relationship between the soil water content and critical height of collapsing
wall, with the critical height of the wall being 8-9 m when the water content is low, which is only
2-3 m in the saturated state. Zhang et al. (2013) pointed out that the granite soil is easy to
disintegrate with increasing water content, and the process is irreversible. Zhang et al.(2012)
proposed that the cohesion and internal friction angle of the soil showed a nonlinear attenuation
trend with the increase of water content, and the shear strength index showed a peak value when
the soil water content was about 13%. Liu et al. (2015) and Deng et al. (2015) reported that the
water content of the collapsing wall gradually increased with the increase of the soil depth. Deng
et al. (2016) proposed that the soil water characteristic curve of granite weathering layer is
different, and the lower soil layers have greater dewatering ability than the upper soil layers. From
these studies, we can find the soil water content is a common influencing factor, and the stability
of the collapsing wall will vary with it. Wang et al. (2000) believe that the mechanical properties
of soil will change significantly when the rain is in full contact with the soil. Similar conclusions
were reported by Luk et al. (1997a) who revealed the main cause for collapse occurrence is the
short-term rainfall intensity. The liquid limit and plastic limit of soil, namely the soil Atterberg
limits, are its highest and lowest water content in the plastic state, which are of important
significance in predicting the influence of surface runoff and rainfall on the collapsing gully. In
recent years, few studies have been performed on the relationship between Atterberg limits and
soil profiles in the hilly granitic region of southern China.

In this paper, we selected four collapsing gullies in the four counties located in a different

latitude of South China to analyze the influence factors for collapsing gully and the relationships
between soil Atterberg limits and soil physico-chemical properties. The objectives of this study
are: 1) to evaluate the similarities and differences in soil Atterberg limits and soil physico-
chemical properties of different weathering profiles among the four collapsing gullies; 2) to
investigate the relationship between soil Atterberg limits and soil physico-chemical properties by
analyzing the status and variation of soil Atterberg limits and 3) to explore the possibility of using
soil Atterberg limits as an integrated index for quantifying collapsing gully and soil weathering
degree of different weathering profiles in the hilly granitic region.





*Insert*: Figure 1.

2     **Materials and methods**
**2.1 Study area**

The sampling plots(22 ˚58′ -29 ˚24′ N ,110 ˚51′ -118 ˚17′ ) are located in the hilly granitic region

of South China, including Tongcheng county in Hubei province, Gan county in Jiangxi province,
Anxi county in Fujian province, Wuhua county in Guangdong province and Cangwu county in
Guangxi province, which are the most serious collapsing gully erosion centers in South China and
thus were selected as the study sites. These study areas are in a temperate monsoonal continental
climate zone, with an average temperature of 15-22℃, an average annual precipitation of about
1500 mm with high variability. The region is dominated by the granite red soil (Humic Acrisols)
and developed in the Yanshan period. The soil erosion is serious in this region, especially the huge
amount of collapsing gullies. There were 1102, 4138, 4744, 22117 and 1592 collapsing gullies in
the Tongcheng county, Gan county, Anxi county, Wuhua county and Cangwu county, respectively.
**2.2 Soil sampling**

According to previous studies and the soil color and soil structural characteristics, the weathering

profiles of the collapsing gullies of the study area in the hilly granitic region can be subdivided into
four soil layers: surface layer, red soil layer, sandy soil layer, detritus layer (Luk et al., 1997a; Zhang
et al., 2012; Xia et al., 2015). Each soil layers have some common characteristics as reported by
Luk et al. (1997a).

This study focused on the development of four collapsing gullies in the south of China, including

Tongcheng county (TC), Gan county (GX), Anxi county (AX) and Wuhua county (WH), where the
development of collapsing gullies is concentrated. The soil samples were collected in surface layer,
red soil layer, sandy soil layer, detritus layer. According to the height of the collapsing gully wall,
we collected 6, 8, 8 and 8 soil samples in four weathered layers, respectively. The detritus layer of
the collapsing gully in Tongcheng County was not exposed, so the soil samples were not collected.
Descriptions of soil sample site and soil sampling depth are given in Tables 1 and 2.

When collecting the samples of each soil layer, about 1-2 kg soil sample was obtained by means

of quartering and transported to the laboratory for measurement of soil Atterberg limits (including
plastic limit and liquid limit) and soil physico-chemical properties (including soil particle density,



organic matter, cation exchange capacity and free iron oxide). At each layer, six soil samples were
obtained by using cutting ring to determine soil bulk density and calculate the total porosity.
**2.3 Soil analysis**
The soil samples were air-dried and then sieved at the fraction <0.452 mm for Atterberg limits
determination, and at <2mm for measurement of soil physical and chemical properties including
particle density, particle-size distribution and chemical analyses. Soil Atterberg limits (liquid
limit, and plastic limit) were determined using the air-dried soil for each layer according to the
standard methods reported in S.I.S.S (1997) after ASTM D 4318-10e1 (2010), i.e. (Stanchi et al.,
2015). The plasticity index and the liquidity index are obtained by the following Eq (1, 2).

Plasticity index= liquid limit- plastic limit          (1)

Liquidity index= (WC $_{insitu}$ - plastic limit) / (liquid limit- plastic limit)    (2)

where WC$_{insitu}$ is in situ water content.
The particle density (PD) was measured by the pycnometer method, the bulk density (BD) was
determined by the cutting ring method, and the total porosity (TP) was calculated as TP = 1 - (BD
/ PD) (Anderson and Ingram, 1993; Cerdà and Doerr, 2010). The particle-size distribution (PSD)
was determined by the sieve and pipette method (Gee and Bauder, 1986). Soil organic matter
(SOM) was measured by the $K_2Cr_2O_7$-$H_2SO_4$ oxidation method of Walkey-Black (Nelson and
Sommers, 1982; Armo et al., 2014). Cation exchange capacity (CEC) was measured after
extraction with ammonium acetate (Rhoades, 1982); Free iron oxide (Fed) were extracted by
dithionite-citrate-bicarbonate (DCB) (Mehra and Jackson, 1958).
**2.4 Statistical analysis**
Statistical analyses were performed by SPSS 19.0 software (SPSS Inc., Chicago, IL, USA). A
one-way analysis of variance (ANOVA) was performed to examine the effects of soil depth on
soil Atterberg limits and soil physico-chemical properties. The least square difference (LSD) test
(at P<0.05) was used to compare means of soil variables when the results of ANOVA were
significant at P<0.05. Regression analysis was used to analyze the relationship between soil
Atterberg limits and soil physico-chemical properties.
**3    Results and discussion**
**3.1 Soil physico-chemical properties**

 



The soil physical and chemical properties for the different weathering profiles in the four collapsing gullies (TC, GX, AX and WH) were described in terms of soil particle density (PD), soil bulk density (BD), total porosity (TP), soil organic matter (SOM), cation exchange capacity (CEC), free iron oxide ($Fe_d$) and particle size distribution (PSD). The values for these properties are shown in Table 2 and Table 3. Average values at varying soil layers including surface soil layer, red soil layer, sandy soil layer and detritus layer are given in Figure 2 and Figure 3.

### 3.1.1 Soil particle density (PD)

From Table 2, it can be seen that the soil PD was the highest in TC3 (2.68 g cm$^{-3}$), GX6 (2.69 g cm$^{-3}$), AX3 (2.66 g cm$^{-3}$) and WH3 (2.72 g cm$^{-3}$) of each collapsing gully, but the lowest in TC1 (2.58 g cm$^{-3}$), GX1 (2.57 g cm$^{-3}$), AX8 (2.53 g cm$^{-3}$) and WH1 (2.52 g cm$^{-3}$). Significant differences (p<0.05) were observed in the average PD values of the different soil layers in TC, GX, AX and WH (Figure 2 A). The PD was the least in the surface soil layer, followed by the detritus layer, which may be related to the higher humus content of the surface soil layer and the looser structure of detritus layer. In addition, the highest PD was observed in the red soil layer of TC, AX and WH and the sandy soil layer of GX, probably due to the large amounts of iron oxide and other heavy minerals they contain. Furthermore, as shown in Table 2, most of the soil PD values in all the four soil layers were less than 2.65 g cm$^{-3}$, which are often used to calculate the value of soil BD (Lee et al., 2009; Sharma and Bora PK, 2015). The lower PD value may be due to the loose structure of granite soil (Luk et al, 1997a).

### 3.1.2 Bulk density (BD)

From Table 2, it can also be seen that soil BD values were the lowest in the surface layer of all the collapsing gullies (1.29 g cm$^{-3}$, 1.27g cm$^{-3}$, 1.21 g cm$^{-3}$ and 1.33 g cm$^{-3}$for TC, GX, AX and WH, respectively). However, relatively higher BD values were observed in the red soil layer (1.47 g cm$^{-3}$, 1.42 g cm$^{-3}$, 1.43 g cm$^{-3}$ and 1.48 g cm$^{-3}$ for TC, GX, AX and WH, respectively), followed by the sandy layer. The average soil BD values had significant difference ($p < 0.01$) in the different soil layers of TC, GX, AX and WH except in the surface layer of WH (Figure 2 B).Meanwhile, the bulk density first increased sharply ($p < 0.01$) and then declined slightly from the surface layer to the sandy soil layer of TC and to the detritus layer of GX, AX and WH (Table 2), which are similar to the report by Perrin et al. (2014). The soil BD values of the surface layer




were lower than those of the other layers, probably due to the higher content of SOM, more plant
root distribution, better soil structure and texture (Choudhury et al., 2015). With the leaching and
deposition process of the surface soil, the fine particles migrated to the red soil layer, leading to
the filling of soil large pores and the increase of soil BD (Huang et al., 2014; Masto et al., 2015).
The lower soil BD values of the sandy layer and detritus layer may be due to weak weathering and
loose soil structure (Lan et al., 2013).

**3.1.3 Total porosity (TP)**

Unlike soil BD, the soil TP was comparatively high in the surface soil layer of GX and WH, but
was the highest in the red soil layer of AX (Figure 2C). From Table 2, it can be seen that the soil
TP values were lower in the red soil layer, such as the TC2 (44.11 %) and GX4 (46.02 %), which
may be due to the weathering process of these soil layers, feldspar and mica in mineralized
granites (Wang et al., 2015; Deng et al., 2016).

**3.1.4 Soil organic matter (SOM)**

Soil organic matter (SOM) plays an important role in soil nutrient availability, its increase may
decrease the potential of soil erosion (Oliveira et al., 2015). As shown in Table 2, with the increase
of depth, SOM contents in the soil layers of the four collapsing gullies showed a sharply
decreasing trend ($P<0.05$). The sandy soil layers and detritus layers showed relatively lower SOM
contents than those in the red soil layers and surface layers (Figure 2D). The AX1 had the highest
SOM content (44.06 g kg$^{-1}$), followed by TC1 (23.37 g kg$^{-1}$), WH1 (15.17 g kg$^{-1}$) and AX2
(11.23 g kg$^{-1}$) (Table 2), which is mainly due to the decomposition of surface litter in the ground
surface. However, the sandy soil layer and the detritus layer are basically in the state of
incomplete weathering, and there is no accumulation of SOM (Xia et al., 2015).

**3.1.5 Cation exchange capacity (CEC)**

Cationic exchange capacity (CEC) is a measure of the soil capacity to adsorb and release
cations (Jordán et al., 2009; Khaledian et al., 2016; Muñoz-Rojas et al., 2016). Similar to the SOM
trend, CEC also decreased significantly from the upper soil layer to the bottom layer in the four
collapsing gullies in TC, GX, AX and WH. As shown in Table 2, the CEC values were the highest
in the surface soil layer of all the collapsing gullies (1.29 g cm$^{-3}$, 1.27g cm$^{-3}$, 1.21 g cm$^{-3}$ and 1.33
g cm$^{-3}$for TC1, GX1, AX1 and WH1, respectively). The average CEC values in the four



collapsing gullies followed the order of surface soil layer> red soil layer> sandy soil layer>
detritus layer with significant difference ($P<0.05$) (Figure 2E).

### 3.1.6 Free iron oxide ($Fe_d$)

$Fe_d$ is the secondary product formed by the weathering of the parent rock during soil formation.
One $Fe_d$ state of the film surface is wrapped in the shape of clay minerals, and another state may
be filled in the micropores of clay minerals (Cerdàet al., 2002; Lan et al., 2013). It is a unique and
very important cementing material in weathered soil. As shown in Table 2, $Fe_d$ values were the
lowest in the detritus layer of all the collapsing gullies (11.89 g kg$^{-1}$, 9.41 g kg$^{-1}$, 7.30 g kg$^{-1}$ and
8.37 g kg$^{-1}$ for TC, GX, AX and WH, respectively). The highest $Fe_d$ values of AX and WH were
observed in the surface soil layer (31.03 g kg$^{-1}$ and 28.40 g kg$^{-1}$ for AX and WH), while those of
TC and GX were observed in the red soil layer (27.37 g kg$^{-1}$ and 26.59 g kg$^{-1}$ for TC and GX).
Overall, there are significant differences among surface soil layer, red soil layer, sandy soil layer
and detritus layer in different weathering profiles (Figure 2F).These results show that the
structural and mechanical properties are stronger in the surface soil layers and the red soil layers.
However, when compared to the upper soil layers, the soil structure is loose and cohesive strength
is low in the sandy soil layer and detritus layer.

### 3.1.7 Particle size distribution (PSD)

Soil particle size distribution (PSD) is one of the most important physical attributes in soil
systems (Hillel, 1980). PSD affects the movement and retention of water, solutes, heat, and air,
and thus greatly affects soil properties (Arjmand Sajjadi et al., 2014). The highest clay contents
were 41.03, 36.65, 53.27 and 32.62% in TC, GX, AX and WH, respectively, and silt varied from
25.67 to 38.21% in TC, 28.43 to 38.68% in GX, 21.06 to 36.75% in AX and 26.90 to 41.51% in
WH. The averages of particle size distributions for different weathering profiles of the four
collapsing gullies are shown in Figure 3. The results indicated that the finer soil particles declined
and the coarse soil particles increased from surface layer to detritus layer. The surface layer of TC,
GX and WH collapsing gullies had the greatest clay content of 32.81, 36.65 and 32.62%,
respectively, while the red soil layer of the AX collapsing gully showed the greatest clay content
(45.63%). The reason for this phenomenon is the different weathering degree of granite, the grain
size becomes coarser, the $SiO_2$ content and sand content increase, and the clay content decreases





from top to the bottom (Xu, 1996; Lin et al., 2015).
*Insert*: Table 2; Table 3 and Figure 2; Figure 3.
**3.2 Soil Atterberg limits characteristics of weathering profiles of the collapsing gullies**
All the measured soil plastic limit and liquid limit values varied significantly among the
different soil layers in the four collapsing gullies (TC, GX, AX and WH). Table 4 lists the
calculated values for the Atterberg limits, plasticity index and liquidity index. The average values
for these properties are shown in Figure 4 and the relationship of these values with soil depth are
shown in Figure 5.
**3.2.1 Soil plastic limit and liquid limit**
As shown in Table 4, soil plastic limit and liquid limit varied greatly from top to the bottom of
different soil layers. Specifically, the soil plastic limit ranged from 19.43 % (TC6) to 35.93 % (TC1)
with an average of 28.34 % in TC, 19.51 % (GX6) to 33.82 % (GX1) with an average of 24.19 %
in GX, 19.32 % (AX7) to 36.03 % (AX2) with an average of 26.87 % in AX, and 18.91 % (WH8)
to 36.56 % (WH8) with an average of 23.98 % in WH. Consistent with the variation trend of plastic
limit, the soil liquid limit was found to be highest in TC1 (62.68 %), GX1 (57.70 %), AX1 (65.71 %)
and WH1 (62.70 %) in each weathering profile of the four collapsing gullies, and lowest in TC6
(30.91 %), GX6 (30.89 %), AX8 (32.48 %) and WH7 (30.77 %). The averages of soil plastic limit
and liquid limit are shown in Table 4. The results indicated that, with declining weathering degree
(from surface layer to detritus layer), the plastic limit and liquid limit decreased noticeably ($p<0.05$)
(Figure 5A; 7B). The surface layer of all the four collapsing gullies had the greatest soil Atterberg
limits (35.93, 33.82, 35.58 and 36.56 % for the plastic limit, and 62.68, 57.70, 65.71 and 62.70 %
for the liquid limit, respectively). The plastic limit of the sandy soil layer and the detritus layer was
significantly lower ($p<0.01$) than that of the surface soil layer and the red soil layer, but with no
significant difference between each other. As shown in Figure 5, the soil Atterberg limits presented
a nonlinear relationship with soil depth. Power function fitting showed that both the soil plastic limit
and liquid limit had a remarkable negative correlation with the soil depth (Figure 5A, $R^2=0.784$,
$p<0.001$ and Figure 5B, $R^2=0.877$, $p<0.0001$, respectively). Additionally, the soil plastic limit of
the surface soil layer and the red soil layer ranged between 24.70 % and 36.56 % with an average
of 31.98 % and the liquid limit ranged between 49.43 % and 65.71 % with an average of 57.02 %,

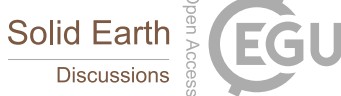

which are higher compared with most types of soil (Reznik, 2016), but an opposite trend was
observed in the sandy soil layer and the detritus layer. The soil plastic limit and liquid limit are
respectively the minimum water content and the maximum water content of the soil in the plastic
state, which reflect the strength of the connection between soil particles and the resistance ability of
the soil to the deformation caused by the external force when the water content is different (Institute
of Soil Science, Chinese Academy of Sciences, 1978). Our findings are in agreement with the
previous studies by Zhuang et al. (2014) and Xia et al. (2016), which reported the upper soil layer
has a better ability to resist deformation than the bottom layer. These results indicate that the change
of water content has little influence on the surface soil layer and the red soil layer, and the soil
cannot be easily transformed into a liquid state by the rainfall erosion and runoff scouring.
Conversely, the change of water content has a great influence on the sandy soil layer and the detritus
layer, and with water content increasing, the soil can be changed from solid to liquid state.
**3.2.2 Soil plasticity index and liquidity index**
Soil plasticity index is an indicator for the difference between liquid limit and plastic limit, while
liquidity index represents the ratio of the difference of the natural moisture content and the plastic
limit to the plastic limit (Zhuang et al., 2014). These indexes were calculated by formulae (1) and
(2). As shown in Table 4, there are considerable differences in soil plasticity index and liquidity
index among the different weathering profiles of the four collapsing gullies. The soil plasticity index
was highest in AX1 (30.14 %), followed by TC1 (26.75 %), GX3 (26.50 %) and WH2 (26.19 %),
and it also was the highest in each soil layer. However, the plasticity index was lowest in the bottom
soil layers (11.48, 10.09 and 11.53% for TC6, GX8 and AX8, respectively) except for WH.
Additionally, inconsistent with plasticity index, liquidity index was the lowest in the surface soil
layer of each weathering profile (-49.55, -50.36, -64.57 and -65.91 % for TC1, GX1, AX1 and WH1,
respectively). The highest liquidity indexes of TC, GX, AX and WH were -10.57 % in TC6, -17.61 %
in GX8, -12.41 % in AX8 and -11.65 % in WH7, respectively. Figure 4 summarizes the statistics of
soil plasticity index and liquidity index in all of the different weathering profiles of the four
collapsing gullies. Significant differences were observed among the surface soil layer, red soil layer,
sandy soil layer and the detritus layer for all the measured plasticity and liquidity indexes. The results
indicated that the soil plasticity index decreased noticeably with the decline of weathering degree



(from surface layer to detritus layer), which is similar to the variation regularity of plastic limit and
liquid limit.
The surface layer of the TC, AX and WH collapsing gullies had the greatest soil plasticity index
(26.75%, 30.14 % and 26.14 %, respectively), but the greatest plasticity index (23.88%) of the GX
collapsing gully was found in the red soil layer. In contrast with the plasticity index, the liquidity
index was significantly ($p<0.05$) higher in the sandy soil layer and the detritus layer and was the
lowest in the surface soil layer (-49.55 %, -50.36 %, -64.57 % and -65.91 % for TC, GX, AX and
WH, respectively) (Figure 4). Regression analyses were performed to determine the strength of
relationships between the plasticity index, the liquidity index and soil depth (Figure 5). The
nonlinear regression analyses showed that the plasticity index had a remarkable negative correlation
with the soil depth (Figure 5C, $R^2=0.759$, $p<0.0001$). However, there was a significant positive
correlation between the soil liquidity index and the soil depth based on the power function fitting
analysis (Figure 5D, $R^2=0.382$, $p<0.05$).
The differences in soil plasticity index and liquidity index between upper layer and lower layer
may be related to the variation in the dynamics of the soil properties. As previously reported [64],
changes in soil plasticity index and liquidity index depend on soil properties. The plasticity index
reflects the range of the soil water content when the soil is in the plastic state. The size of the
plasticity index is directly related to the maximum possible bound water content of a certain mass
of soil particles. The greater the maximum possible bound water content is, the greater the
plasticity index will be. However, the bound water content of soil is related to the size of soil
particle, mineral composition, the composition and concentration of cation in the hydration
membrane. Thus, the plasticity index is a comprehensive indicator for the reaction properties of
clayey soil, which means the larger the index is, the higher the clay content will be (Husein et al.,
1999). Our findings clearly demonstrated that the plasticity index of lower soil layers was
significantly lower ($p<0.01$) than that of the upper layers in the different weathering profiles of the
four collapsing gullies, implying that the content of fine particles in the soil gradually decreased
with soil depth. Previous studies about soil texture classification are frequently based on soil
plasticity index: the soil with a value between 10% and 17% is defined as silty clay and that with a
value greater than 17% is classified as clay (Zentar et al., 2009; Marek et al., 2015). Therefore,



based on this classification theory, most soil layers in the TC, GX, AX and WH collapsing gullies
can be defined as clay, while the lower soil layers can be classified as silty clay, which is more
susceptible to erosion.
However, the adsorption capacity of bound water varied under a different soil specific surface
area and mineral composition. Therefore, given the same water content, for the soil with high
viscosity, the water may be bound water, while for the soil with low viscosity, a considerable part
of the water can be free water, which means that the soil state cannot be defined only by water
content and we need another indicator, namely the liquidity index, to reflect the relationship
between natural water content and Atterberg limits in the soil. The liquidity index is defined as the
ratio of the difference between the natural moisture content and the plastic limit to the plastic limit
(Sposito, 1989). When the natural moisture content is close to the plastic limit, the soil is hard; and
when it is close to the liquid limit, the soil is weak. In engineering practice, the soil is in a hard
state when the liquidity index is less than 0 (Zhuang et al., 2014). In our research, the liquidity
indexes of all soils were less than 0, indicating that the soil of the different weathering profiles of
the four collapsing gullies is hard in the natural state. Nevertheless, the lower soil layer of the
collapsing gullies is more close to 0 than the upper layer in the liquidity index, indicating that the
lower soil is weaker than the upper layer soil.

*Insert*: Table 4 and Figure 4; Figure 5.

**3.2.3 Relationship between soil Atterberg limits and collapsing gully**
The ability of soil to resist external erosion varies with soil Atterberg limits. In this study, the
liquidity indexes of all soils were less than 0, indicating that the soils of the four collapsing gullies
remain solid in natural state, with a high shear strength and strong resistance to water erosion,
enabling the soil of granite weathering profile to maintain stability. From the soil Atterberg limits
of all the soils of the four collapsing gullies, it can be seen that the plastic limit, liquid limit and
plasticity index are higher in the surface soil layer and red soil layer, implying that the plastic state
cannot be easily changed when the rain lasts a short time such as moderate to light rain, which
usually does not lead to the collapse and loss of the soils with high compaction and hardness.
However, if the rainfall duration continues long enough, the soil water content can reach a high
level, leading to the increase of the soil self-weight, the decrease of the soil shear strength, and then





the collapse of the soils. The plastic limit, liquid limit and plasticity index of the sandy soil layer
and detritus layer of the collapsing gully are significantly smaller than those of the surface soil layer
and red soil layer, indicating that it is very easy for the soils to reach the plastic limit in the case of
short-term rainfall, and coupled with the looser soil and smaller soil shear strength, it is easy for
them to collapse.
Because of the lower soil Atterberg limits of the collapsing gully in the bottom soil layers, soil
moisture absorption leads to the increase of water content after a long time of rain erosion and soil
preferential flow. The sandy soil layer and detritus layer of the collapsing gully would be the first
to reach or close to the plastic state in the same moisture conditions. Meanwhile, the shear strength
of the two soil layers decreased rapidly, leading to the formation of the weak surface and then the
collapse or water erosion. The erosion is much more serious in the sandy soil layer and detritus layer
than in the surface soil layer and red soil layer, resulting in the hollow-out of the lower soil layers
and the formation of a concave pit called "niche" in the engineering geology (Ding et al., 1995;
Deng et al., 2016). The formation and development of the niche is the preliminary stage of the
formation of a collapsing gully. After niche formation, the surface soil layer and red soil layer lack
support, giving rise to a total collapse by the soil self-weight. The occurrence of collapse forms the
source of erosion, resulting in the formation of the collapsing gully.
**3.3 Effect of soil physico-chemical properties on soil Atterberg limits**
In this research, we examined the soil particle density (PD), bulk density (BD), total porosity
(TP), soil organic matter (SOM) , cation exchange capacity (CEC), free iron oxide ($Fe_d$) and
particle size distribution (PSD) among the different soil layers in the four collapsing gullies (TC,
GX, AX and WH). The relationship between soil physico-chemical properties and soil Atterberg
limits are shown in Table 4 and Figure 6.
*Insert*: Table 5 and Figure 6.
**3.3.1 Soil particle density (PD), bulk density (BD) and total porosity (TP)**
Regression analyses were performed to determine the strength of relationships between Atterberg
limits and soil particle density, bulk density and total porosity in the soil of the four collapsing
gullies (TC, GX, AX and WH). In the four collapsing gullies, soil Atterberg limits had a very weak
negative correlation with the soil BD ($R^2= 0.044$, $p<0.05$ for plastic limit; $R^2= 0.021$, $p<0.05$ for





liquid limit) and PD ($R^2$= 0.023, p<0.05 for plastic limit; $R^2$= 0.002, p<0.05 for liquid limit), and a
very weak positive correlation with the soil TP ($R^2$= 0.117, p<0.05 for plastic limit; $R^2$= 0.074,
p<0.05 for liquid limit). Therefore, there was almost no significant correlation between soil
Atterberg limits and PD, BD and TP in the soils of the four collapsing gullies.
**3.3.2 Soil organic matter (SOM)**
In reference to Figure 6, regression analyses showed that the soil organic matter had a
significant positive correlation with plastic limit ($R^2$=0.816, P<0.01) and liquid limit ($R^2$=0.785,
P<0.01) in all of the different weathering profiles of the four collapsing gullies. This may be due
to the reason that soil organic matter can promote organic colloid formation, which can affect the
specific surface area, the water holding capacity of the soil particles and then the soil liquid limit
(Stanchi et al., 2012). With the increase of organic matter content, organic colloid also increased,
indicating/implying that the greater the water holding capacity of the soil is, the greater the liquid
limit will be. In our research, the soil Atterberg limits had a significant positive correlation with
the organic matter in all of the different weathering profiles. Similar results were also reported by
Zhuang et al. (2014) and Husein et al (1999), who both concluded that the plastic limit and the
liquid limit of the soil increase with increasing organic content. According to the relationship
between the Atterberg limits and the organic matter in the weathering profiles of the granite soil,
we can conclude that the higher the content of organic matter is, the stronger the anti-erodibility of
the soil will be. Thus, our research provides a theoretical basis for the prevention and control of
collapsing gully erosion by planting green manure to improve soil organic matter in these areas.
**3.3.3 Cation exchange capacity (CEC)**
As shown in Figure 6, there was a strong positive correlation between soil Atterberg limits and
CEC ($R^2$= 0.636, p<0.01 for plastic limit; $R^2$= 0.739, p<0.01 for liquid limit). Similar results were
reported by Cathy et al. (2008), who put forward that CEC can be an indicator for the mineral type
and is highly correlated to plastic limit and liquid limit.
**3.3.4 Free iron oxide (Fe$_d$)**
A positive significant correlation was observed between soil Atterberg limits and Fed ($R^2$= 0.630,
p<0.01 for plastic limit; $R^2$= 0.788, p<0.01 for liquid limit) (Figure 6). This is consistent with the
finding of Stanchi (2015), who reported that Atterberg limits were also affected by CEC. Therefore,





$Fe_d$ acts as an inorganic binding agent in structure formation, and participates in reducing horizon
vulnerability, as proposed by Sposito (1989).
**3.3.5 Particle size distribution (PSD)**
Regression analyses were performed to determine the strength of relationships between soil
Atterberg limits and the contents of gravel, coarse sand, fine sand, silt and clay in the soils of
collapsing gullies (Figure 6). The non-linear regression analyses showed a strong positive
correlation of the soil Atterberg limits with the clay content ($R^2= 0.736$, $p<0.01$ for plastic limit;
$R^2= 0.820$, $p<0.01$ for liquid limit), a remarkable negative correlation with the content of sand
($R^2= 0.580$, $p<0.01$ for plastic limit; $R^2= 0.616$, $p<0.01$ for liquid limit) and a weak negative
correlation with the silt content ($R^2= 0.320$, $p<0.05$ for plastic limit; $R^2= 0.210$, $p<0.05$ for liquid
limit), gravel content ($R^2= 0.255$, $p<0.05$ for plastic limit; $R^2= 0.202$, $p<0.05$ for liquid limit),
coarse sand content ($R^2= 0.214$, $p<0.05$ for plastic limit; $R^2= 0.374$, $p<0.05$ for liquid limit) and
fine sand content ($R^2= 0.131$, $p<0.05$ for plastic limit; $R^2= 0.158$, $p<0.05$ for liquid limit). The
significant negative correlation between soil Atterberg limits and sand may be attributed to
porosity and specific surface area. When the sand content increases, the soil pores will increase
and surface area will decrease, resulting in poor soil performance and facilitating water
movement. Meanwhile, sandy soil is low in viscosity, loose and difficult to expand, leading to the
slow rise of capillary water during water erosion. Therefore, the soil plastic limit and liquid limit
will decrease with increasing sand content. Our results show that with declining weathering degree
(from surface layer to detritus layer), the sand increased and the finer soil particles declined,
which causes the decrease of soil Atterberg limits, and the lower soil layers are the first to be
eroded (Zhuang et al., 2014).
Furthermore, there was a significant positive correlation between soil Atterberg limits and clay
content, indicating that the clay content, despite its modest amount, plays a major role in
determining the values of plastic limit and liquid limit. This also shows that, in the weathering
profiles, the soil Atterberg limits increased with the increase of clay content, which is also reported
by several other studies (Polidori, 2007; Baskan et al., 2009; Keller and Dexter, 2012). This result
may be due to the effect of clay on soil plasticity in changing the arrangement of soil particles. The
connection form, the arrangement of soil particles and soil pore size will vary greatly with the clay



content. Additionally, soil clay has a larger specific surface area, which will affect the soil water

storage capacity. Therefore, the huge specific surface area enables the clay to have strong

adsorption capacity, which affects the speed of water flow in the soil. Meanwhile, the mosaic of

clay particles to the larger pores can also block the flow channels in the soil. All of these will

affect the soil Atterberg limits, with the high clay content contributing to the directional

arrangement of soil particles, leading to the increase of weak bound water content, thereby

increasing the plastic limit and liquid limit of the soil.

Overall, soil is a spheres of the earth system with special structure and function. From the point

of view of the earth's circle, soil science should not only study the soil material, but also should

change towards the relationship between the soil and the earth's circle, which has a profound

impact on human living environment and global change research (Brevik et al., 2015; Keesstra et

al., 2016). The results show that the relationship between soil Atterberg limits and the occurrence

mechanism of collapsing gully, which can be used as a reference for the assessment of natural

disasters occurring in the interaction between water and force in nature.

**4    Conclusions**

Based on the analyses of soil Atterberg limits, soil physico-chemical properties, the influence

factors on collapsing gully and the relationships between soil Atterberg limits and soil physico-

chemical properties of different weathering profiles of the four collapsing gullies in the hilly granitic

region, the conclusions are summarized as follows:

Different weathering profiles exhibit a significant effect on soil Atterberg limits and soil physico-

chemical properties. The upper soil layers (surface layer or red soil layer) of all the collapsing gullies

show the highest plastic limit, liquid limit, plasticity index, SOM, CEC, $Fe_d$, finer soil particles and

the lowest liquidity index, PD, and BD. With the fall of weathering degree (from surface layer to

detritus layer), there is a sharp decrease in the plastic limit, liquid limit, plasticity index, SOM, CEC

and $Fe_d$, a gradual increase in liquidity index, a sharp increase in PD and BD first followed by a

slight decline. Additionally, the finer soil particles (silt and clay) decrease, and especially the clay

contents decline noticeably, whereas the gravel and sand contents increase considerably. Therefore,

the soils of bottom layers are very easy to reach the soil Atterberg limits during rain, and coupled

with the looser soil structure, they are easy to be eroded, resulting in the hollow-out of these soil

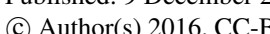



layers and the formation of a concave pit called "niche" in engineering geology. After the niche
formation, the upper soil layers lack support, leading to a total collapse in the soil by the soil self-
weight. The collapse occurrence forms the source of erosion, causing the formation of the collapsing
gully. The regression analysis shows that soil Atterberg limits are significantly positively correlated
with SOM, clay content, CEC and $Fe_d$, remarkably negatively correlated with sand content and not
obviously correlated with other properties. The results of this study demonstrate that soil Atterberg
limits can be regarded as an informative indicator to reflect the weathering degree of different
weathering profiles of the collapsing gully. Future research will include the relationship between
soil Atterberg limits and soil mechanical properties.

*Author contributions*. Conceived and designed the experiments: Y. S. Deng, C. F. Cai and J. Z.
Chen. Performed the experiments: Y. S. Deng and D. Xia. Analyzed the data: Y. S. Deng.
Contributed reagents/materials/analysis tools: Y. S. Deng, D. Xia and S. W. Ding. Wrote the
paper: Y. S. Deng, C. F. Cai, D. Xia, S. W. Ding and J. Z. Chen.

*Acknowledgements*. Financial support for this research was provided by the National Natural
Science Foundation of China (No.41630858; 41601287 and 41571258) and National Science and
technology basic work project (No.2014 FY110200A16). We would like to thank several
anonymous reviewers for their valuable comments on the previous version of the manuscript.
Finally, thanks to all of our colleagues who supported the undertaking of this work.

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

**Tables**
**Table 1.** Description of soil sample site
**Table 2.** Description of weathering profile, soil sampling depth and soil properties for different weathering profiles of the four
collapsing gullies
**Table 3.** Percentages of different particle-size distributions for different weathering profiles of the four collapsing gullies





**Table 4.** Soil Atterberg limits for different weathering profiles of the four collapsing gullies

**Table 5.** Regression and correlation analysis of soil Atterberg limits with soil physico-chemical properties

**Table 1.** Description of soil sample site

| Location | Collapsing gully code | Longitude and latitude | Altitude (m) | Height of collapsing gully wall(m) | Coverage of tree layer (%) | Coverage of surface layer (%) | Vegetation community |
|---|---|---|---|---|---|---|---|
| Tongcheng County | TC | 29°12′39″N 113°46′26″E | 142 | 9 | 45 | 64 | *Pinus massoniana + Cunninghamia lanceolata + Liquidambar formosana + Phyllostachys heterocycla - Rosa laevigata + Smilax china + Gardenia jasminoides + Vaccinium carlesii + Lespedeza bicolor - Dicranopteris linearis + Miscanthus floridulus* |
| Gan County | GX | 26°11′22.2″N 115°10′39.4″E | 175 | 15 | 35 | 38 | *P. massoniana + L. formosana + Schima superba - L. bicolor - D. linearis* |
| Anxi County | AX | 24°57′14.3″N 118°3′35.1″E | 172 | 20 | 30 | 43 | *P. massoniana + Eucalyptus robusta + Acacia confusa - Rhus chinensis + Rhodomyrtus tomentosa + Loropetalum chinense - D. linearis +M. floridulus* |
| Wuhua County | WH | 24°06′10.4″N 115°34′57.1″E | 157 | 35 | 28 | 35 | *P. massoniana - R. tomentosa + Baeckea frutescens - D. linearis* |

**Table 2.** Description of weathering profile, soil sampling depth and soil properties for different weathering profiles of the four collapsing gullies

| Soil layer code | Weathering profile | D (m) | PD (g cm$^{-3}$) | BD (g cm$^{-3}$) | TP (%) | SOM (g kg$^{-1}$) | CEC (cmol kg$^{-1}$) | Fed (g kg$^{-1}$) |
|---|---|---|---|---|---|---|---|---|
| TC1 | Surface layer | 0.3 | 2.58 | 1.29 ±0.05d | 49.03 ±2.37a | 23.37 ±0.55a | 16.39 ±0.90a | 21.38 ±0.46bc |
| TC2 | Red soil layer | 0.8 | 2.64 | 1.47 ±0.01a | 44.11 ±0.29c | 6.81 ±0.17b | 8.37 ±1.14b | 27.37 ±0.84a |
| TC3 | Red soil layer | 2 | 2.68 | 1.34 ±0.05c | 49.53 ±1.79a | 5.84 ±0.20c | 7.59 ±0.27b | 23.29 ±1.29b |
| TC4 | Red soil layer | 4 | 2.65 | 1.39 ±0.02b | 47.26 ±0.85b | 2.68 ±0.13d | 3.32 ±0.44c | 19.42 ±1.72c |
| TC5 | Sandy soil layer | 7 | 2.62 | 1.33 ±0.02c | 49.72 ±0.83a | 1.20 ±0.11e | 4.07 ±0.61c | 13.84 ±0.93d |
| TC6 | Sandy soil layer | 9 | 2.65 | 1.35 ±0.01c | 48.63 ±0.35ab | 1.02 ±0.06e | 3.92 ±0.34c | 11.89 ±1.00e |
| | | | | | | | | |
| GX1 | Surface layer | 0.3 | 2.57 | 1.27 ±0.05c | 50.94 ±2.34a | 7.93 ±0.11a | 10.28 ±0.17a | 25.31 ±1.45a |
| GX2 | Red soil layer | 0.8 | 2.67 | 1.40 ±0.03ab | 47.65 ±1.50b | 1.35 ±0.08b | 8.27 ±0.44bc | 26.59 ±2.90a |
| GX3 | Red soil layer | 1.8 | 2.64 | 1.40 ±0.02ab | 46.79 ±0.87bc | 1.07 ±0.12c | 7.91 ±0.60c | 22.72 ±0.57bc |
| GX4 | Red soil layer | 4 | 2.63 | 1.42 ±0.02a | 46.02 ±0.95c | 0.86 ±0.07d | 8.90 ±0.69b | 23.96 ±1.11b |
| GX5 | Sandy soil layer | 7.5 | 2.62 | 1.41 ±0.02ab | 46.13 ±1.06c | 0.42 ±0.06f | 5.41 ±0.86d | 18.36 ±0.77c |
| GX6 | Sandy soil layer | 9 | 2.69 | 1.37 ±0.04bc | 49.20 ±1.59ab | 0.72 ±0.09e | 5.98 ±0.52d | 13.30 ±0.43d |
| GX7 | Detritus layer | 11 | 2.64 | 1.33 ±0.06c | 48.32 ±1.27b | 0.40 ±0.06f | 2.09 ±0.19e | 9.90 ±0.78e |
| GX8 | Detritus layer | 13.5 | 2.59 | 1.38 ±0.04ab | 46.65 ±1.96bc | 0.71 ±0.11e | 3.43 ±0.36e | 9.41 ±0.63e |
| | | | | | | | | |
| AX1 | Surface layer | 0.3 | 2.54 | 1.31 ±0.06c | 44.40 ±2.78d | 44.06 ±0.04a | 22.18 ±0.21a | 31.03 ±1.80a |





| AX2 | Red soil layer | 0.8 | 2.63 | 1.39 ±0.06ab | 54.24±2.89a | 11.23 ±0.61b | 14.63±1.30b | 27.53±0.56b |
| AX3 | Red soil layer | 2 | 2.66 | 1.43 ±0.03a | 52.38±1.73ab | 6.33 ±0.11c | 9.20±0.58c | 26.35±0.74b |
| AX4 | Red soil layer | 4 | 2.60 | 1.41 ±0.01a | 50.81±0.45b | 2.41 ±0.11d | 6.37±0.61d | 24.38±1.11c |
| AX5 | Sandy soil layer | 8 | 2.65 | 1.37 ±0.03b | 48.39±1.31bc | 0.82 ±0.03f | 4.82±0.18e | 11.87±1.04d |
| AX6 | Sandy soil layer | 10 | 2.54 | 1.35 ±0.02bc | 47.01±0.88c | 1.31 ±0.09e | 5.02±0.27de | 10.55±1.23d |
| AX7 | Detritus layer | 12 | 2.62 | 1.32 ±0.02c | 49.50±0.82bc | 0.81 ±0.07f | 2.36±0.32f | 7.34±0.56e |
| AX8 | Detritus layer | 15 | 2.53 | 1.31 ±0.02c | 48.12±1.33bc | 0.67 ±0.09f | 3.80±0.71ef | 7.30±0.80e |
| | | | | | | | | |
| WH1 | Surface layer | 0.3 | 2.52 | 1.33 ±0.04d | 48.19±0.93a | 15.17 ±1.73a | 13.84±0.88a | 28.40±0.64a |
| WH2 | Red soil layer | 1 | 2.69 | 1.48 ±0.01b | 44.96±0.29c | 4.65 ±0.29b | 7.69±0.39b | 24.52±0.54b |
| WH3 | Red soil layer | 2.5 | 2.72 | 1.47 ±0.03b | 45.68±1.15bc | 2.59 ±0.14c | 6.62±0.51b | 22.94±0.91bc |
| WH4 | Sandy soil layer | 5 | 2.68 | 1.44 ±0.02c | 46.15±0.83b | 2.82 ±0.03c | 6.54±0.45b | 16.28±1.10c |
| WH5 | Sandy soil layer | 9 | 2.63 | 1.40 ±0.03cd | 46.44±1.64b | 1.61 ±0.10d | 4.18±0.50c | 12.41±0.27d |
| WH6 | Sandy soil layer | 11 | 2.62 | 1.49 ±0.02b | 43.01±1.01c | 0.57 ±0.08f | 2.28±0.22d | 14.23±0.78cd |
| WH7 | Detritus layer | 14 | 2.59 | 1.54 ±0.03a | 40.34±1.46d | 0.74 ±0.05e | 3.91±0.18cd | 8.86±0.40e |
| WH8 | Detritus layer | 17 | 2.61 | 1.37 ±0.05d | 46.41±1.59b | 0.23 ±0.18g | 1.93±0.30e | 8.37±0.32e |

Values with different letters are significantly different at the P < 0.05 level among the different soil layers of the same collapsing
gully. SOM: soil organic matter; $Fe_d$ =Free iron oxide
**Table 3.** Percentages of different particle-size distributions for different weathering profiles of the four collapsing gullies

| Soil layer code | Mass percentages of soil particle-size distribution (mm) | | | | | | | | |
|---|---|---|---|---|---|---|---|---|---|
| | Gravel | Coarse sand | | Fine sand | | Silt | | | | Clay |
| | 2.0-12.0-1.0 | 1.0-0.5 | 0.5-0.25 | 0.25-0.15 | 0.15-0.05 | 0.05-0.02 | 0.02-0.01 | 0.01-0.005 | 0.005-0.002 | <0.002 |
| TC1 | 9.24±1.61b | 7.13±0.10d | 7.09±1.35b | 3.97±0.64d | 9.86±0.93c | 6.55±1.67d | 12.07±0.59a | 5.16±0.58c | 6.11±0.81b | 32.81±1.46b |
| TC2 | 7.87±0.65b | 6.55±0.12e | 6.12±0.54c | 6.10±0.07c | 6.24±0.93d | 16.67±1.04a | 9.81±0.50b | 6.18±1.07b | 5.54±0.92c | 28.91±0.62c |
| TC3 | 4.51±0.36c | 4.91±0.24f | 5.27±0.11d | 6.72±0.85bc | 10.55±1.14c | 6.34±1.22d | 9.74±1.16b | 3.66±0.84d | 7.26±0.21a | 41.03±0.72a |
| TC4 | 3.05±0.55d | 7.95±0.54c | 9.78±1.08a | 9.19±1.32a | 17.66±1.57a | 6.25±0.60d | 10.97±0.96a | 3.27±0.63d | 5.69±0.55c | 26.19±1.86d |
| TC5 | 5.34±0.71c | 11.14±0.38b | 11.75±0.78a | 10.21±1.05a | 13.68±1.45b | 14.01±1.16b | 9.44±0.17b | 7.54±0.25a | 6.64±0.79b | 10.24±0.18e |
| TC6 | 19.84±2.28a | 14.63±0.58a | 11.95±1.23a | 7.58±0.37b | 16.46±1.04a | 8.28±0.91c | 8.48±0.98c | 5.20±0.33c | 3.71±0.13d | 3.87±0.48f |
| | | | | | | | | | | |
| GX1 | 8.99±0.37d | 4.78±0.10d | 4.43±0.29e | 3.94±0.18e | 12.77±0.34f | 2.92±0.25e | 5.49±0.78d | 6.09±1.03e | 13.92±1.65a | 36.65±1.85a |
| GX2 | 8.12±0.31e | 4.66±0.19d | 4.41±0.05e | 4.17±0.22e | 13.62±0.31de | 4.14±0.66d | 7.92±1.27bc | 7.00±1.10d | 12.85±1.62a | 33.10±1.80b |
| GX3 | 9.89±0.50c | 5.65±0.21c | 6.19±0.25d | 5.32±0.41d | 16.40±1.03c | 9.24±0.33c | 7.19±1.74c | 8.50±0.65a | 10.37±0.88b | 21.25±1.14c |
| GX4 | 8.85±0.71d | 5.68±0.30c | 7.93±0.31b | 8.68±0.53b | 18.72±1.27b | 8.80±0.45c | 8.09±0.21b | 7.65±0.48c | 9.81±0.41bc | 15.78±0.39d |
| GX5 | 9.71±1.30cd | 5.03±0.25d | 4.17±0.39e | 4.91±0.42d | 27.91±0.96a | 11.14±0.54b | 8.49±1.4b | 6.68±1.43c | 7.69±1.25d | 14.29±0.55d |
| GX6 | 12.13±0.73b | 7.90±0.19b | 7.30±0.19c | 8.69±0.40b | 16.40±0.34c | 12.44±0.52a | 8.62±0.59b | 8.24±0.53a | 9.37±0.71c | 8.90±0.42f |
| GX7 | 14.87±1.28a | 8.87±0.14a | 8.60±0.81ab | 9.84±0.99a | 14.60±0.72d | 10.37±1.63bc | 6.03±0.82d | 8.83±0.17a | 4.44±1.99e | 13.55±1.39de |
| GX8 | 15.83±0.85a | 8.80±0.07a | 8.67±0.20a | 8.09±0.62c | 13.15±0.99ef | 11.18±1.11ab | 9.73±1.47a | 7.68±0.31c | 5.31±1.46e | 11.55±1.11e |
| | | | | | | | | | | |
| AX1 | 19.32±0.48c | 7.55±0.42c | 6.67±0.23c | 3.86±0.18d | 6.52±0.94d | 5.04±0.95d | 6.02±0.37d | 3.63±0.47e | 7.93±0.24c | 33.47±1.39b |
| AX2 | 6.23±0.35e | 5.34±0.16d | 4.10±0.31d | 2.90±0.23ef | 4.42±0.33e | 3.47±0.71e | 4.01±0.19e | 6.34±1.12c | 11.53±1.90ab | 51.66±1.54a |



| Soil layer code | | | | | | | | | |
|---|---|---|---|---|---|---|---|---|---|
| AX3 | 6.39±0.25e | 5.66±0.21d | 3.99±0.43d | 3.21±0.13e | 6.42±1.02d | 4.19±0.97de | 1.60±0.62f | 5.64±1.35cd | 9.61±0.69b | 53.27±1.47a |
| AX4 | 8.65±0.74d | 4.63±0.08e | 3.31±0.16e | 2.48±0.50f | 12.22±1.02c | 3.92±1.81e | 8.27±1.17ab | 11.65±0.56a | 12.91±1.91a | 31.96±0.55b |
| AX5 | 19.86±0.87bc | 8.71±0.23b | 6.08±0.29c | 5.35±0.12c | 14.30±1.81bc | 8.62±0.48c | 8.02±1.53b | 8.35±0.37b | 4.04±1.32d | 16.68±1.10c |
| AX6 | 24.49±1.05a | 10.01±0.42a | 7.66±0.45b | 6.44±1.02ab | 15.82±1.44ab | 10.71±0.50b | 6.87±1.11cd | 6.58±1.13c | 4.27±0.07d | 7.14±1.33d |
| AX7 | 19.15±0.35c | 7.83±0.27c | 7.04±0.57b | 5.95±0.69b | 15.96±0.78a | 15.85±1.12a | 8.00±0.74bc | 8.00±0.48b | 3.78±0.73d | 8.45±0.31d |
| AX8 | 21.02±1.37b | 10.93±0.43a | 10.86±0.98a | 7.94±1.76a | 17.48±1.97a | 8.73±1.08c | 9.00±0.30a | 5.01±0.27d | 1.02±0.49e | 8.00±1.25d |
| | | | | | | | | | | |
| WH1 | 18.53±0.62f | 5.67±0.12c | 3.74±0.17c | 2.30±0.39d | 10.24±1.15a | 9.33±1.30a | 5.55±0.19d | 4.59±0.62d | 7.42±1.85d | 32.62±1.30a |
| WH2 | 23.42±0.40d | 5.78±0.09c | 2.93±0.21de | 2.29±0.05d | 6.89±0.74c | 7.34±0.56c | 8.51±1.28a | 3.70±0.55d | 10.23±1.32c | 28.92±2.22b |
| WH3 | 25.72±1.91b | 5.92±0.29c | 2.76±0.08e | 1.97±0.05d | 5.15±0.18d | 5.74±0.53d | 4.29±0.63c | 8.72±0.93c | 12.91±0.15b | 26.83±1.82b |
| WH4 | 22.26±1.33de | 6.39±0.21b | 3.24±0.25d | 2.06±0.10d | 4.96±1.10d | 5.45±1.25d | 7.09±1.00bc | 9.10±0.60c | 16.07±1.60a | 23.38±1.97c |
| WH5 | 24.53±0.62c | 8.46±0.16a | 4.29±0.27b | 3.05±0.14c | 5.67±1.34d | 7.02±0.76c | 4.04±0.94e | 15.15±1.85a | 10.23±1.03c | 17.54±1.67d |
| WH6 | 27.73±0.23a | 8.50±0.41a | 5.00±0.49a | 4.40±0.37b | 3.06±0.38e | 10.94±1.25a | 6.98±1.34bc | 12.39±0.65b | 10.06±1.73c | 10.93±1.38e |
| WH7 | 25.81±0.25b | 8.54±0.05a | 5.29±0.29a | 5.57±0.24a | 9.27±0.86ab | 8.36±1.80ab | 6.73±0.73c | 14.46±1.25ab | 5.56±0.38d | 10.42±0.79e |
| WH8 | 25.16±0.82b | 8.48±0.17a | 5.42±0.08a | 5.24±0.61a | 8.43±0.49b | 7.40±1.66bc | 7.55±1.80ab | 15.65±1.21a | 10.91±0.57c | 5.77±0.82f |

Values with different letters are significantly different at the $P < 0.05$ level among the different soil layers of the same collapsing
gully.

**Table 4.** Soil Atterberg limits for different weathering profiles of the four collapsing gullies

| Soil layer code | Plastic limit (%) | Liquid limit (%) | Plasticity index (%) | Liquidity index (%) |
|---|---|---|---|---|
| TC1 | 35.93±0.69a | 62.68±1.32a | 26.75±2.01a | -49.55±3.74d |
| TC2 | 31.73±2.25b | 53.09±0.20bc | 21.36±2.05b | -47.08±4.52d |
| TC3 | 30.51±0.72b | 56.03±2.20b | 25.52±1.47a | -27.60±1.59b |
| TC4 | 31.74±0.56b | 50.04±0.23c | 18.30±0.33c | -35.54±6.96c |
| TC5 | 20.73±1.68c | 35.31±1.05d | 14.58±2.73d | -37.25±6.96c |
| TC6 | 19.43±2.07c | 30.91±0.25d | 11.48±1.82d | -10.57±1.68a |
| | | | | |
| GX1 | 33.82±0.13a | 57.70±2.16a | 23.88±2.04ab | -50.36±4.29e |
| GX2 | 27.04±2.81b | 52.91±0.61b | 25.87±2.20a | -34.67±2.94d |
| GX3 | 23.08±0.45c | 49.58±0.96bc | 26.50±1.41a | -30.54±1.62c |
| GX4 | 23.97±2.39c | 45.82±3.61c | 21.85±1.22b | -25.80±1.44bc |
| GX5 | 22.88±1.98cd | 43.32±1.45c | 20.44±0.53b | -24.27±0.63bc |
| GX6 | 19.51±0.95d | 30.89±2.02e | 11.38±1.07d | -22.42±2.10b |
| GX7 | 21.16±1.53cd | 34.25±0.41d | 13.09±1.12c | -18.16±1.57a |
| GX8 | 22.06±0.59cd | 32.15±1.44de | 10.09±2.03d | -17.61±3.56a |
| | | | | |
| AX1 | 35.58±1.70a | 65.71±0.02a | 30.14±1.72a | -64.57±3.70d |
| AX2 | 36.03±2.83a | 60.67±0.11ab | 24.64±2.72b | -52.16±5.76c |
| AX3 | 35.42±0.21a | 57.01±4.56b | 21.59±4.36bc | -52.00±10.49c |
| AX4 | 25.84±1.60b | 48.34±0.71c | 22.49±2.31bc | -26.59±2.73b |




| | | | | |
|---|---|---|---|---|
| AX5 | 22.34±1.65bc | 40.66±0.12cd | 18.32±1.53c | -24.12±2.00b |
| AX6 | 19.51±0.44d | 32.51±1.18e | 13.00±0.74d | -24.27±1.40b |
| AX7 | 19.32±0.31d | 36.26±0.98d | 16.94±0.68cd | -13.35±0.54a |
| AX8 | 20.95±1.36c | 32.48±1.36e | 11.53±0.02e | -12.41±0.01a |
| | | | | |
| WH1 | 36.56±0.99a | 62.70±1.04a | 26.14±0.05a | -65.91±0.13e |
| WH2 | 26.01±2.36b | 52.20±0.97b | 26.19±3.32a | -31.84±4.03b |
| WH3 | 24.93±0.17bc | 46.86±2.09c | 21.93±1.92b | -42.67±3.74d |
| WH4 | 23.83±0.10c | 46.11±0.86c | 22.28±0.96b | -38.60±1.68bcd |
| WH5 | 22.25±0.62c | 39.11±0.29d | 16.87±0.33c | -36.69±0.70bc |
| WH6 | 19.74±0.84d | 34.22±1.95e | 14.48±1.11cd | -13.38±1.00a |
| WH7 | 19.56±0.27d | 30.77±1.32f | 11.21±1.59d | -11.65±1.63a |
| WH8 | 18.91±1.44d | 31.72±0.48f | 12.81±1.93d | -12.24±1.85a |


**Table 5.** Regression and correlation analysis of soil Atterberg limits with soil physico-chemical properties

| | Plastic limit | | Liquid limit | |
|---|---|---|---|---|
| | Regression equations | $R^2$ | Regression equations | $R^2$ |
| Gravel content | $y = -5.083\ln(x) + 38.722$ | 0.255 | $y = -8.323\ln(x) + 66.423$ | 0.202 |
| Coarse sand content | $y = -8.895\ln(x) + 48.448$ | 0.214 | $y = -21.66\ln(x) + 100.51$ | 0.374 |
| Fine sand content | $y = -4.772\ln(x) + 38.804$ | 0.131 | $y = -9.633\ln(x) + 71.562$ | 0.158 |
| Sand content | $y = -17.16\ln(x) + 90.809$ | 0.580 | $y = -32.52\ln(x) + 168.51$ | 0.616 |
| Silt content | $y = -19.2\ln(x) + 91.772$ | 0.320 | $y = -28.59\ln(x) + 143.51$ | 0.210 |
| Clay content | $y = 7.6773\ln(x) + 3.4506$ | 0.736 | $y = 14.915\ln(x) + 1.8834$ | 0.820 |
| BD | $y = -28.04\ln(x) + 34.789$ | 0.044 | $y = -35.65\ln(x) + 56.651$ | 0.021 |
| PD | $y = -49.17\ln(x) + 73.088$ | 0.023 | $y = -27.35\ln(x) + 71.436$ | 0.002 |
| TP | $y = 35.364\ln(x) - 110.82$ | 0.117 | $y = 51.702\ln(x) - 154.49$ | 0.074 |
| SOM | $y = 4.2553\ln(x) + 22.753$ | 0.816 | $y = 7.6856\ln(x) + 39.781$ | 0.785 |
| CEC | $y = 7.9009\ln(x) + 11.719$ | 0.636 | $y = 15.682\ln(x) + 17.359$ | 0.739 |
| $Fe_d$ | $y = 10.629\ln(x) - 4.226$ | 0.630 | $y = 21.885\ln(x) - 16.509$ | 0.788 |


**Figure Captions**
**Figure 1**. A typical collapsing gully in the hilly granitic region, Anxi County, Fujian Province
**Figure 2.** Average of soil properties for different weathering profiles of the four collapsing gullies.
**Figure 3.** Average of different particle-size distributions for different weathering profiles of the four collapsing gullies.
**Figure 4.** Average of soil Atterberg limits for different weathering profiles of the four collapsing gullies.
**Figure 5.** Relationship between soil Atterberg limits and soil depth.



**Figure 6.** Correlations between soil Atterberg limits and soil physico-chemical properties.

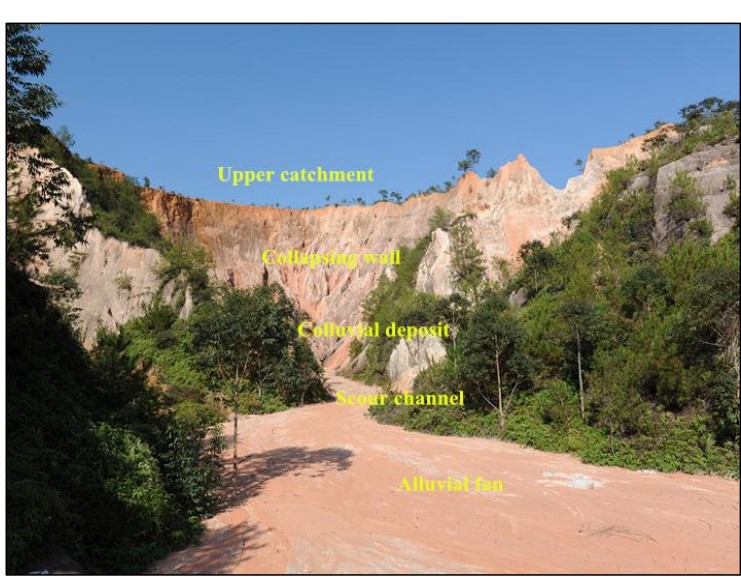


**Figure 1.** A typical collapsing gully in the hilly granitic region, Anxi County, Fujian Province (photo: Shuwen Ding)

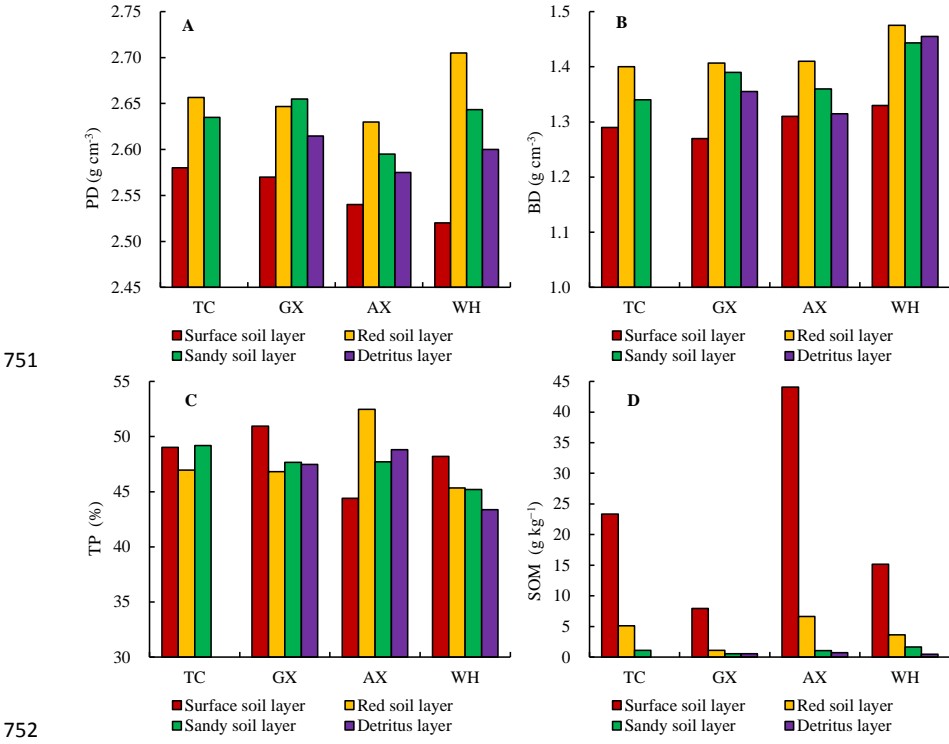





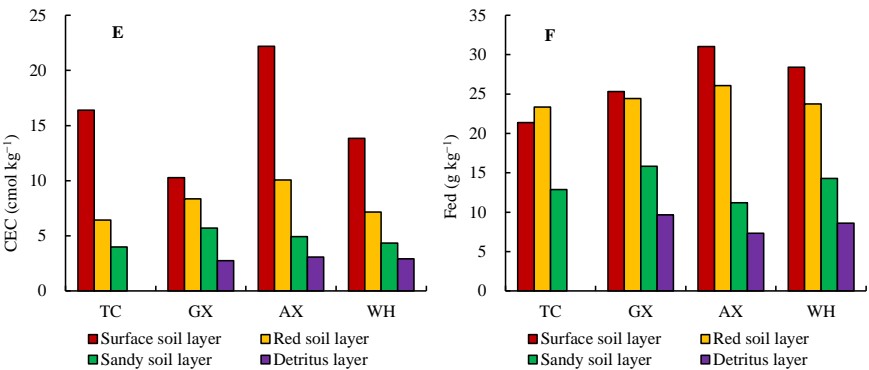

**Figure 2.** Average of soil properties for different weathering profiles of the four collapsing gullies. (A) particle density; (B) bulk

density; (C) total porosity; (D) soil organic matter; (E) cation exchange capacity; and (F) Free iron oxide.

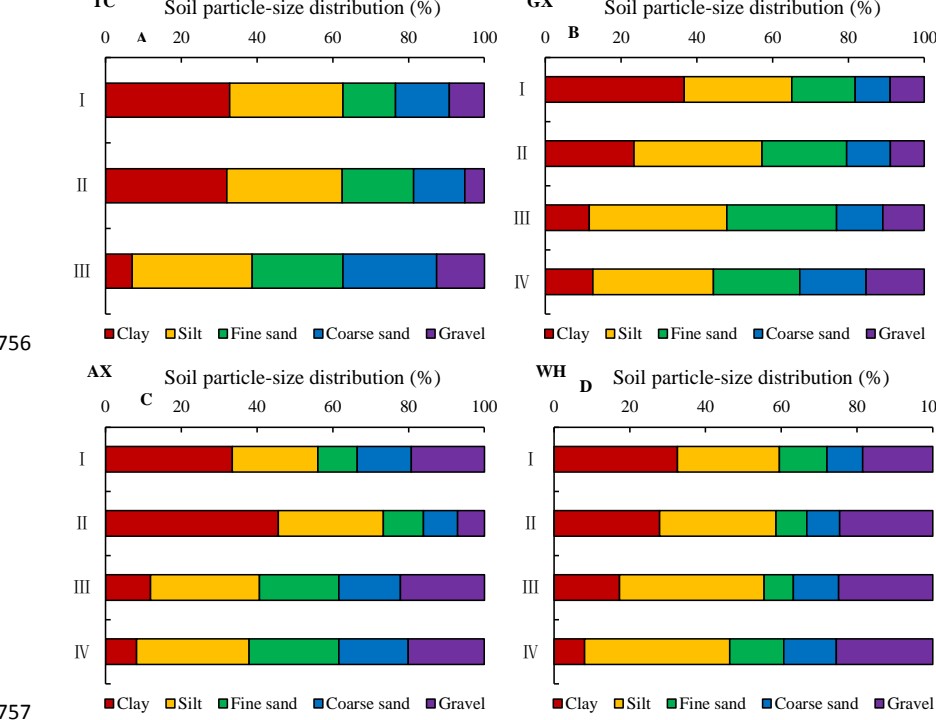

Ⅰ: Surface soil layer; Ⅱ: Red soil layer; Ⅲ: Sandy soil layer; Ⅳ: Detritus layer

**Figure 3.** Average of different particle-size distributions for different weathering profiles of the four collapsing gullies. (A)

Tongcheng county; (B) Ganxian county; (C) Anxi county; and (D) Wuhua county.





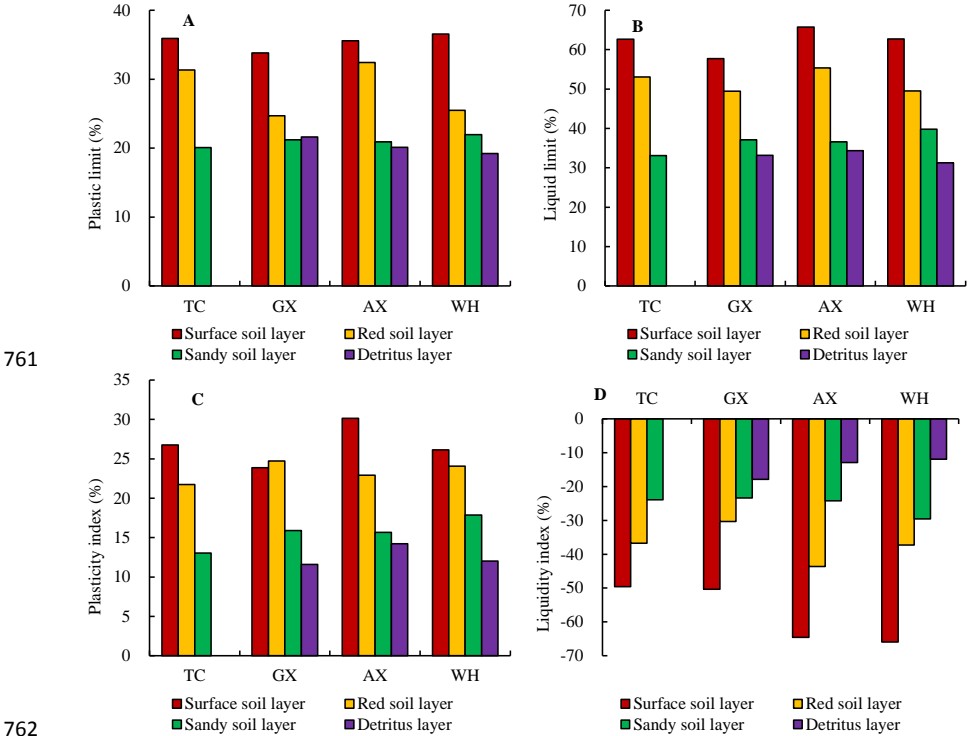

**Figure 4.** Average of soil Atterberg limits for different weathering profiles of the four collapsing gullies. (A) plastic limit; (B) liquid limit; (C) plasticity index; and (D) liquidity index.

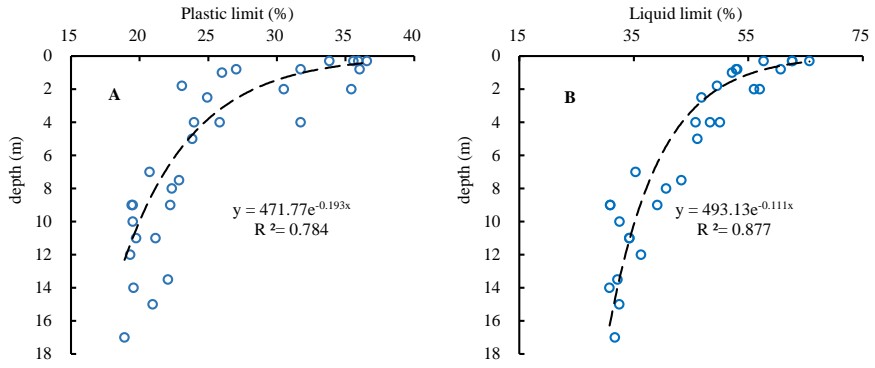





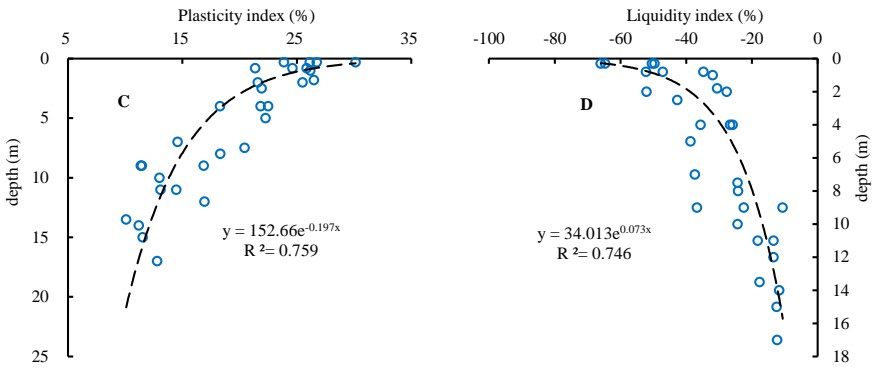

**Figure 5.** Relationship between soil Atterberg limits and soil depth. (A) plastic limit; (B) liquid limit; (C) plasticity index; and (D) liquidity index.

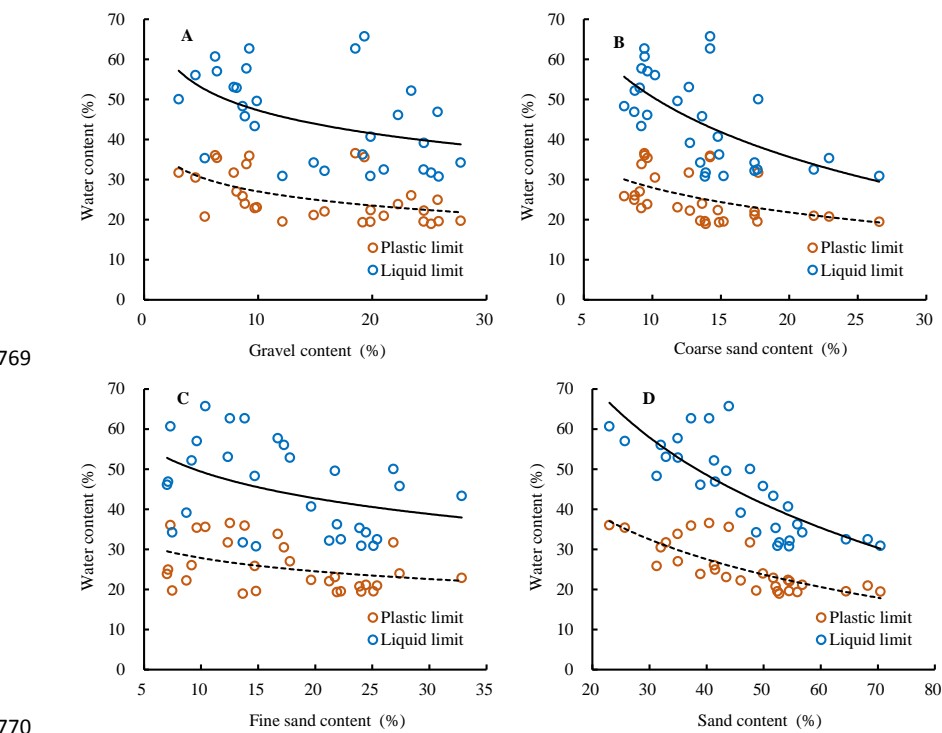








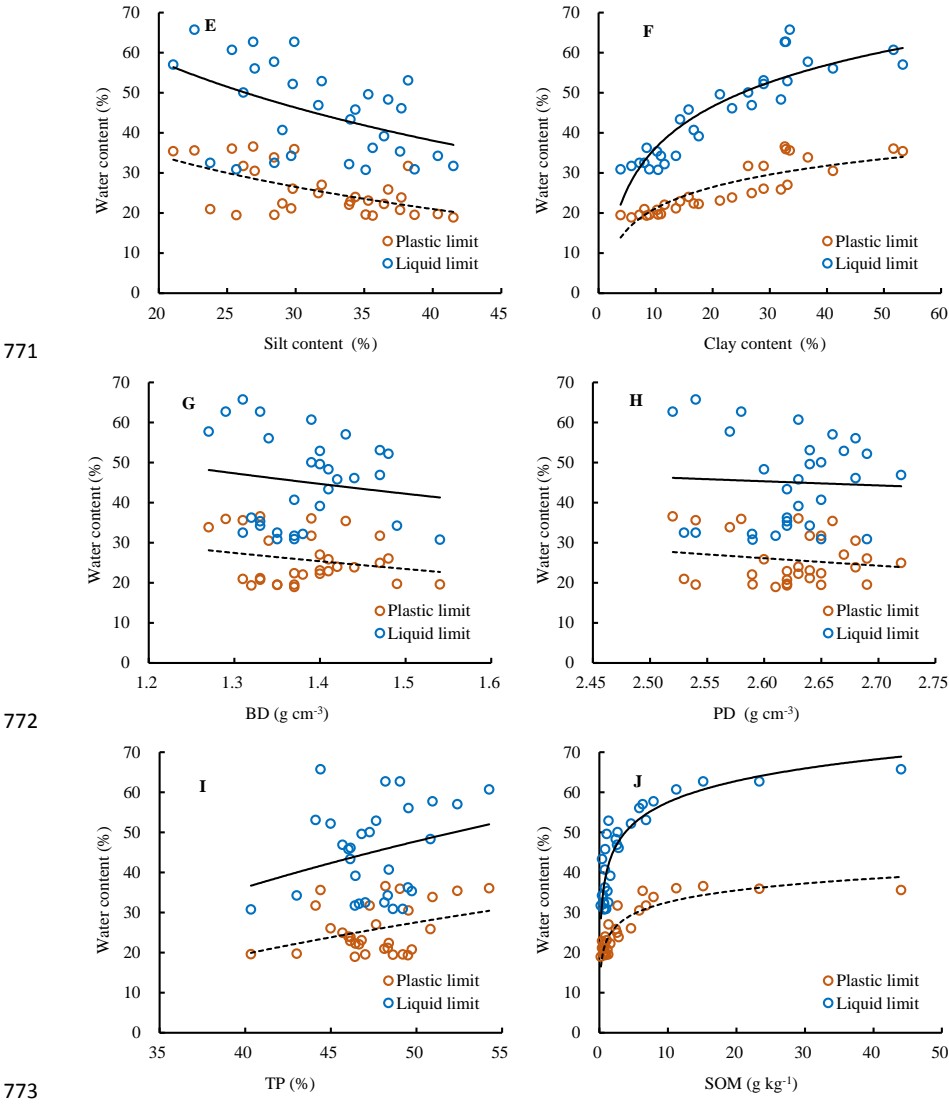



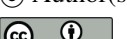

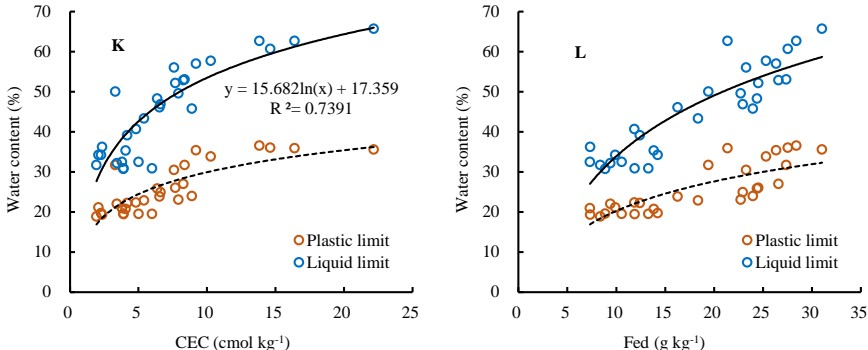


**Figure 6.** Correlations between soil Atterberg limits and soil physico-chemical properties. (A) gravel content; (B) Coarse sand

content; (C) fine sand content; (D) sand content; (E) silt content; (F) clay content; (G) bulk density; (H) particle density; (I) total

porosity; (J) soil organic matter; (K) cation exchange capacity; and (L) Free iron oxide.
