# Peer review of "Soil Atterberg limits of different weathering profiles of the collapsing 1 gullies in the hilly granitic region of south China 2 3 Yusong Deng 1, Chongfa Cai 1\*, Dong Xia 2, Shuwen Ding 1, Jiazhou Chen 1 4 5 ${}^{1}Key\ Laboratory\ of\ Arable\ Land\ Conservation\ (Middle\ and\ Lower\ Reaches\ of\"

_Solid Earth, 2016_

## Referee Comment (RC1) · F. Pacheco (Referee) · 24 Dec 2016

REVISION Paper: se-2016-152 Title: Soil Atterberg limits of different weathering profiles of the collapsing gullies in the hilly granitic region of south China Corresponding author: Chongfa Cai Reviewer: Fernando A.L. Pacheco

OUTLINE AND GENERAL APPRECIATION This study presents an assessment of soil Atterberg limits in weathering profiles of four collapsing gullies located in a hilly region of South China, relating them to various soil physic-chemical properties. Atterberg limits are shown to vary along the profiles, being followed by concomitant variations of some soil properties (e.g., positive correlation with SOM, clay content, CEC; negative correlation with sand content). The study is very interesting, being well written and

organized. It is also well documented and merits publication in Solid Earth as it fits well in the scope of this journal. However, a fundamental issue needs to be addressed before the manuscript is ready for publication. See details below.

CONCERNS There are no control sites in this study, which is a serious weakness of this study. To be used as indicators of collapsing gullies, the characteristics of weathering profiles (Atterberg limits, soil properties) need to be assessed in control hillsides as well, i.e. hillsides not affected by collapsing gullies. Only the characteristics that are found statistically different between the unaffected and affected hillsides can be used as indicators. This has not been checked by the authors and must be accomplished in the revised version. In its present form, the Atterberg and/or soil properties profiles and can eventually be used to interpret the causes of collapsing gullies, but are not able to indicate a profile to distinguish among gully-affected and gully-unaffected hillsides.

RECOMMENDATION Major revision 24 December 2016

Kind regards, Fernando A.L. Pacheco

---

## Referee Comment (RC2) · Anonymous Referee #2 · 29 Dec 2016

Submitted to:
**Solid Earth**

**General Comments:**
This article discusses the use of Atterberg limits as an indicator for soil vulnerability and degradation in collapsing gully systems in the hilly granitic regions of south China. The relationship of the limits to certain soil-physico-chemical characteristics and different weathering profiles was also explored. Overall, the manuscript is sufficiently novel and significant. The authors have in particular done a thorough job of highlighting the impact of collapsing gully systems to physical and social environments in the region. The article is well referenced and concepts explained sufficiently. However, there are a few minor issues related to sampling and site selection (see below under specific comments).

**Specific Comments:**
The authors have used appropriate physico-chemical soil measurements with which to compare the limits and weathering profiles. However, all the sites that were selected from already collapsing gullies with no unaffected site with which to compare to. As such, the relationships studied can be related to the Atterberg limits and weathering profiles to some extent but not to a true baseline of conditions of an unaffected slope. This study would be considerably strengthened if this was included in the results and subsequent statistical analyses.

**Technical corrections:**
There are a few minor technical/typographical corrections listed below:

| Page # | Line # | |
|--------|--------|--------------------------------------------------------------------------------------|
| 3 | 80 | Change sentence to **Soil erosion is an important problem in mountainous areas**… |
| 3 | 82 | Be careful where you mention collapsing gully, check for the correct use of singular/plural (also |
| 4 | 105-106 | check for in other parts of the text) |
| 4 | 98-99 | Perhaps think of another way of framing this sentence |
| 4 | 113 | Change sentence to **there is a close relationship between  the stability of…** |
| 5 | 119 | Change sentence to **pointed out that  granite soil is easy to…** |
| 6 | 148 | Add space between **plots (22** |
| 6 | 161-163 | Consider adding a photo figure here to illustrate the four soil layers in more detail. |
| 8 | 229 | Add space between **B). Meanwhile** |

**Decision:**
Revise (minor corrections)

---

## Referee Comment (RC3) · Anonymous Referee #3 · 3 Jan 2017

REVISION Paper: se-2016-152 Title: Soil Atterberg limits of different weathering profiles of the collapsing gullies in the hilly granitic region of south China

GENERAL COMMENTS: This study discusses the use of soil Atterberg limits in weathering profiles of four collapsing gullies located in a hilly región of South China, as an indicator for soil vulnerability and degradation in collapsing gully systems, relating them to various soil physic-chemical properties. The paper, well written and organized, is very interesting. It is also well referenced. However, in my opinion there are a few doubts related to sampling and site selection to be resolved before publishing.

SPECIFIC COMMENTS: I miss control sites that allow to use Atterberg limits or soil properties as indicators of collapsing gullies. Characteristics of weathering profiles

should be compared with unaltered slopes. This would modify the statistical study.

RECOMMENDATION: Minor Revision

---

## Referee Comment (RC4) · Anonymous Referee #4 · 4 Jan 2017

General comment:

The study falls within the scope of Solid Earth. The paper is interesting, but it is not performed to be well understood. There are numerous amendments and explanations required. The English is not understandable in many places. The manuscript is too wordy and contain repeated facts already previously mentioned (see specific comments). Also, there is lack of novelty of this study. Although phenomenon of collapsing gully deserves better explanation of indicators, present study does not explain it. This is main fault of the paper. Without comparison of soil properties with close surrounding soil not affected with by collaps, present paper only repeats well known facts of correlation of Atterberg limits with presented soil properties. When updating, please compare

your results with surrounding soil. Also, take more deep insight in relation: type of clay vs. dominating exchangeable cations (in CEC). This can help to explain better the results of plastic and liquid limit in each location. Some statements and conclusions are well known from earlier literature, while part of section are beyond the limits of the results (conclusion in lines 519-525). For these reasons I can not recommend this paper for publication in this condition.

Some specific comments are below:

Line 37-42: I notice that authors use long sentences. In many occasions long sentences make paper difficult to follow. Please rather use shorter and direct statements. Line 47: "…leading to a loss of bottom soil layer…" Unclear, please rewrite. Line 62-65: This statement is unclear. What are cited authors find? This is example of absence of continuity in paper. In several occasions I missed the point I was following. Line 65-67: Please add word "recommended". This sentence can mean that you cannot till wet soil. It is not recommended if you want to keep your soil structural stability. Line 73-75: Please state sources in end of sentence. Line 77-79: Avoid general statements. It is better to state an important finding of each study you cite. Also, please avoid adjectives like "mountain soils". Soils have their own characteristics. Line 80-82: Please recheck English again. Line 98-100: Unclear sentence. Please rewrite. Line 105-110: Authors repeat again previously mentioned facts. Delete it. Line 119: Define "granite soil". Line 125-126: unify words, eg. weathering layer – upper soil layers. Use "topsoil" and "subsoil layers" instead. Line 131-133: Please delete "The liquid limit and plastic limit of soil, namely..". It is already mentioned before. Line 133-135: Authors write general statement. Please avoid this. Great part of Introduction authors explain the influence of water content on collapsing gully, but here is not mentioned the existing research about relation of Attenberg limits vs. soils characteristics in hilly southern China. Please write the papers who studied similar problem and their key findings. Here you can explain the novelty of your paper. Line 136-137: this goes in Materials and methods section. Line 155: Please find a better word than "serious". Line 164-166. Repeating informations mentioned before. Please delete. Line 167-168: It is unclear how many samples authors have: per layer, per location and in total. Please state this informations. Line 194-200. Did authors use appropriate statistical test? Is your data normal? Please test it and state the test and p value. Also, authors in whole manuscript repeat "regressions" but I can find only correlations between paired soil characteristics. Authors can also test regressions between 3 and more data and see the nature of their connections. Line 212-214: This is huge shortcoming. Please upload a statistics in Figure 2, 4 and 5. Line 2014-2016: This is speculation. This paper did present aggregate distribution and humus content. Add source that confirm this finding or delete it. Line 216-218: Content of iron oxide does not correspond with highest PD at each location. Line 284-288: This is contrary with explanation of BD in line 233-235. Line 295-297: Missing statistics in Fig 4 and 5. Line 332-335: Do not repeat facts mentioned before (methods section).Please delete it. Line 343 and 356: Please, add letter beyond number of figure. Line 362: Change number into reference. Line 362-363: Which soil properties? Can authors draw a comparison with their own results? Line 363-367: Instead the authors explain again what plasticity index mean, it is better to explain their relationship with other soil properties and try to compare them with different studies. Line 396: Please define "external erosion" Line 428: Change Table 4 into Table 5 Line 433-437: Authors stated the significant relationships, although some of them are very weak. Please recheck your statistics again.

Sections 3.3.: Sections that contain correlations between variables are well known fact previously published (eg. particle size distributions vs Attenberg limits). Instead try to find differences between collapsing gullies and surrounding soils and then find a "go home message". Line 500-506: Delete it. Conclusions: Rewrite according the suggestions from general comments. Figures: Please delete figure 6. The same data exist in table 5. Add statistics in tables 2, 4 and 5. Define relationship between variables used for correlations in Materials and Methods section. Tables: Add p value in table 5.

---

## Author Comment (AC1) · 17 Jan 2017

Thank you very much for your comments and suggestions. We all agree with you on "the Atterberg and/or soil properties profiles and can eventually be used to interpret the causes of collapsing gullies, but are not able to indicate a profile to distinguish among gully-affected and gully-unaffected hillsides". In fact, collapsing gullies are affected by many factors, climate, topography, soil, etc. But this paper only describes the relationship between soil properties and collapsing gully. We will further explore other factors that cause the collapsing gully. In addition, we will try our best to add control sites in the revised version. Thanks again!

---

## Author Comment (AC2) · 17 Jan 2017

Thank you very much for your recognition and recommendation on this paper. We will try our best to add control sites in the revised version. At the same time, we also enhance the data collation and analysis. Thanks again!
* * *

---

## Author Comment (AC3) · 17 Jan 2017

Thank you very much for your recognition and recommendation on this paper. We will try our best to add control sites in the revised version. At the same time, we also enhance the data collation and analysis. We will correct many details of the writing. Thanks again!

———————————————

---

## Author Comment (AC4) · 17 Jan 2017

Thank you for your very careful advice on this paper. These comments are very useful to us. We will carefully revised in the manuscript. Including research content and data processing, as well as English expression problems. Thank you again!
* * *